# Electrical Excitation Decay Time in Chains of Nanoscale Non-Point Dipoles

**DOI:** 10.3390/nano11010074

**Published:** 2020-12-31

**Authors:** Evgeny G. Fateev

**Affiliations:** Udmurt Federal Research Center, Ural Branch RAS, 426067 Izhevsk, Russia; e.g.fateev@gmail.com

**Keywords:** nanoscale, soft-colloids, self-assembly, decay time, electric excitations

## Abstract

On the basis of a previously developed model of disperse systems containing non-point dipole particles self-assembled into chains inside a liquid substrate, the decay time of electrical excitations induced in dipoles by an external field is investigated. It was shown that when the external field is completely turned off (from $10^{-6}$ V / m to $10^{6}$ V / m levels) at biologically significant low frequencies (for example, 13 Hz), the decay time of the excitations of nanoscale dipoles nonlinearly depends on the chain length. It was found that the decay time of excitations increases sharply (by four to five orders of magnitude), with an increase in the chain length more than 19–20 dipoles.

## 1. Introduction

In response to the action of low-frequency electric fields [1,2,3] nanoscale colloidal spherical particles (with radii r=10−100 nm) have the property of self-assembly in chains [4,5,6], rings and other configurations [7]. Particles with other geometric shapes can also have the property of self-assembly [8]. This property of nanoscale particles is widely used in photonics [9], biosensorics [10], electronics [11] and other high technologies [12,13]. However, it is still unclear how long the structural formations such as chains of dipoles can remain stable in a material substrate [14] or a biological medium [15] after the action of the alternating field upon them is over [16,17,18].

The objective of the present paper is to demonstrate the existence of the excitation decay in the dipole chains using numerical calculations. We will also show the dependence of the excitation decay time on the length and other parameters of the chain.

## 2. Model

The dipole chains can be quickly formed from individual polar colloidal particles [19,20] with induced variable moments under the action of external electric fields at low frequencies. The most suitable model equations for the physics of the chains of dipoles of the colloidal type under the action of external low-frequency $\Omega <10^{5}$ Hz electric fields were presented in [21,22,23].

In the model calculation, the approximation of short-range interaction between the charged nanoscale particles was taken [24]. However, the critical parameters of the model [25] were necessarily taken into account such as the polarizability of the electric double layers of colloidal particles [26], dielectric permeability of the surrounding medium and also the correspondence to the Coulomb law of the reciprocal interaction of the charges on the dipole poles [22]. The above-mentioned approach [22] allows to find a very high probability of the existence of supersensitive responses in the described types of chains to super-weak periodic excitation electric fields. The use of the above approach in some previous investigations allowed to find the dependences of the responses on the length of the chains and the time of the action of low-frequency electric fields upon the chains. Further, we will first show that when the external excitation field is switched off, the internal electric excitations in the chains do not cease all at once but gradually. The degree of the excitations decay depends on the chain length and other parameters.

Let us study the decay of the excitations in the chain of non-point dipoles with the radius *r* and the minimal spacing a→2r as shown in Figure 1.

Let us calculate the interactions between the dipoles in the Coulomb approximation. Let the angles $\varphi_{n-1}$, $\varphi_{n}$, $\varphi_{n+1}$ of the corresponding dipole oscillators 1–3 characterize the deflections of the dipole axes from the positions of unstable equilibrium (see Figure 1). Then the combined view of the potential energy of the chain with dipole–dipole interactions can be written in the form [22]:(1)$$\begin{array}{c} U_{iint}=\frac{1}{4\pi \varepsilon \varepsilon_{0}}\sum_{n}\left\{\frac{Q_{n-1}^{+}Q_{n}^{+}R_{n-1,n}^{++}}{{\left(R_{n-1,n}^{++}\right)}^{2}}+\frac{Q_{n-1}^{-}Q_{n}^{-}R_{n-1,n}^{--}}{{\left(R_{n-1,n}^{--}\right)}^{2}}\right.\\ -\frac{Q_{n-1}^{-}Q_{n}^{+}R_{n-1,n}^{-+}}{{\left(R_{n-1,n}^{-+}\right)}^{2}}-\frac{Q_{n-1}^{+}Q_{n}^{-}R_{n-1,n}^{+-}}{{\left(R_{n-1,n}^{+-}\right)}^{2}}\\ +\frac{Q_{n}^{+}Q_{n+1}^{+}R_{n,n+1}^{++}}{{\left(R_{n,n+1}^{++}\right)}^{2}}+\frac{Q_{n}^{-}Q_{n+1}^{-}R_{n,n+1}^{--}}{{\left(R_{n,n+1}^{--}\right)}^{2}}\\ -\frac{Q_{n}^{-}Q_{n+1}^{+}R_{n,n+1}^{-+}}{{\left(R_{n,n+1}^{-+}\right)}^{2}}-\frac{Q_{n}^{+}Q_{n+1}^{-}R_{n,n+1}^{+-}}{{\left(R_{n,n+1}^{+-}\right)}^{2}}\}. \end{array}$$

Here, $R_{n-1,n}^{++}\;,\;R_{n-1,n}^{--}$, $\;R_{n-1,n}^{-+},\;R_{n-1,n}^{+-}$, $\;R_{n,n+1}^{++},\;R_{n,n+1}^{--}$, $R_{n,n+1}^{-+},\;R_{n,n+1}^{+-}$ are radius-vectors between the corresponding charges in the dipole chain; ε is the dielectric permeability of the medium between the particles, and $\varepsilon {}_{0}$ is the dielectric constant.

For building a one-dimensional model of the system of non-point dipole oscillators with variable moments it is necessary to take into consideration that the values of the charges in the sheath of the dipole *n* should depend on the strength of external and local fields created by moving adjacent charges in the shells of the dipoles n−1 and n+1. Let us assume that the influence of non-adjacent dipoles on one another is effectively screened and is mediated only by the chain. On considering the phenomenon of polarization in oscillators rather formally, we ignore all other possible physicochemical processes in them and around them (for example, see [22,23]). Let us subject the contribution of external and local fields into the polarization of any of the charges $Q_{n}^{+}$,$Q_{n}^{-}$,$Q_{n-1}^{+}$,$Q_{n-1}^{-}$ to the principle of superposition taking into account their effective influence depending on frequency. It is sufficient to formally subject the dependence of the polarization on the frequency of the local $\omega_{n}$ or external Ω excitation for a single particle to, for example, Debye dispersion equation [22,23]. Then, for the values of the positive charges (let us take into account that $∥Q_{n}^{+}∥=∥Q_{n}^{-}∥$ ) at the ends of the dipole shells n−1, *n* and n+1 we will write, respectively:(2)$$\begin{array}{c} Q_{n}^{+}=\beta \;\left\{\frac{c_{o}e\left(\varepsilon {\;}_{s}-\varepsilon {\;}_{\infty }\right)R_{n,n-1}^{++}}{4\pi \;\varepsilon \;\varepsilon_{o}\left(1+{\left(\tau \omega_{n-1}\right)}^{2}\right){\left(R_{n,n-1}^{++}\right)}^{3}}\right.\\ +\frac{c_{o}e\left(\varepsilon {\;}_{s}-\varepsilon {\;}_{\infty }\right)R_{n,n+1}^{++}}{4\pi \;\varepsilon \;\varepsilon_{o}\left(1+{\left(\tau \omega_{n+1}\right)}^{2}\right){\left(R_{n,n+1}^{++}\right)}^{3}}\\ -\frac{c_{o}e\left(\varepsilon {\;}_{s}-\varepsilon {\;}_{\infty }\right)R_{n,n-1}^{+-}}{4\pi \;\varepsilon \;\varepsilon_{o}\left(1+{\left(\tau \omega_{n-1}\right)}^{2}\right){\left(R_{n,n-1}^{+-}\right)}^{3}}\\ -\frac{c_{o}e\left(\varepsilon {\;}_{s}-\varepsilon {\;}_{\infty }\right)R_{n,n+1}^{+-}}{4\pi \;\varepsilon \;\varepsilon_{o}\left(1+{\left(\tau \omega_{n+1}\right)}^{2}\right){\left(R_{n,n+1}^{+-}\right)}^{3}}+\left.\frac{E_{n}^{ext}}{1+{\left(\tau \;\Omega \right)}^{2}}\right\}. \end{array}$$

Similar expressions can also be written for the charges $Q_{n-1}^{+}$ and $Q_{n+1}^{+}$. Here, τ is relaxation time for bound charges in the shells, $\varepsilon_{s}$ and $\varepsilon_{\infty }$ are maximal low-frequency and minimal high-frequency values of dielectric permeability, respectively. The value $c_{0}$ is the number of elementary charges at the ends of dipoles which provides the change of the system dielectric permeability per unit in the particles polarization processes. The external uniform harmonic disturbing field (for simplicity, directed perpendicular to the line of the axis of the dipole chain) in the vicinity of the particle *n* can be written in the form:(3)$$E_{{}_{n}}^{ext}=2\varepsilon^{-1}E\sin\left(2\pi \Omega \;t\right)\cos\left(\varphi {\;}_{n}\right).$$

Similar expressions for the disturbing field can be also written for the vicinities of the dipoles n−1 and n+1. During finding the fields $E_{n}$, $E_{n-1}$ and $E_{n+1}$ it is assumed that the charges at the ends of *n* and n−1 dipoles can take the value $Q_{\infty }=c_{o}e\varepsilon_{\infty }$ for $\omega_{n}\to \infty$, and for $\omega_{n}\to 0$ the value $Q_{0}=c_{0}e\left(\varepsilon_{s}-\varepsilon_{\infty }\right)$. We think that for all the particles in the chain $c_{0}=const$ the coefficient of proportionality β corresponds to the value of the charge at the ends of the dipoles induced in the field with the strength of E=1 V/m.

The expression for the kinetic energy of the chain with the charges with the masses $M_{n}=c_{n}m$ concentrated at the ends of the dipoles has the form
(4)$$T_{k}=\frac{1}{2}\sum_{n}J_{n}{\dot{\varphi }}_{n}^{2}$$

Here, $J_{n}=c_{n}mr^{2}$ is the moment of inertia, $c_{n}$ is the number of uncompensated charges (for example, cations or anions) with the mass *m* in the shell of the dipole *n*.

Let us assume that the dissipative forces linearly depend on the angular rate of the motion of the charges. Then, the corresponding dissipative function for the chain with the dissipation parameter $\xi_{n}$ will take the form:(5)$$D=\frac{1}{2}\sum_{n}c_{n}\xi_{n}r^{2}{\dot{\varphi }}_{n}^{2}.$$

For the *n*th dipole, the fraction of its kinetic energy (apart from thermal fluctuations) that is converted into heat is proportional to the dielectric-loss tangent $tg\delta_{n}\approx \omega_{n}\tau$ where $\omega_{n}\tau$ is the natural frequency of the dipole [27]. The energy dissipation of a chain of dipoles will be proportional to the dissipative parameter:ξ∼ωτm.

The force of the interaction of the external field and the chain of oscillators can be written as follows:(6)$$F_{n}=2\varepsilon^{-1}E\sin\left(2\pi \Omega \;t\right)\sum_{n}Q_{n}\cos\left(\varphi {\;}_{n}\right),$$
where the value of the charges at the ends of the dipoles additively depends on the local and external fields according to relation (Equation 2).

Using Euler–Lagrange equation [22,23] taking into account the dissipation (Equation 5) and the external excitation (Equation 6), we find the equation of motion for Lagrangian:(7)$$L=T_{k}-U_{int}.$$

Assuming that at the same instant of time the variables $\varphi_{n}$ for neighboring dipoles differ slightly, i.e., in the continuum approximation $\varphi_{n}-\varphi_{n-1}∼\delta$ when the transition $na\to x,\varphi_{n}\left(t\right)\to \varphi \left(x,\;t\right)$ takes place, we find the following nonlinear equation of motion resembling the sine-Gordon equation [22,23]: (8)$$\frac{\partial^{2}\varphi }{\partial t^{2}}+\upsilon_{o}^{2}\frac{\partial^{2}\varphi }{\partial x^{2}}-\theta_{o}^{2}\sin\left(\varphi \right)-\eta \frac{\partial^{\varphi }}{\partial^{t}}=\gamma \left(x,t\right).$$

The definitions of the parameters $\upsilon_{o}$, η and γ(x,t) are fully presented in works [22,23].

As a result, we find the number of the uncompensated charges $c_{n}$ at the dipole *n* end
(9)$$c_{n}=Q_{n}^{+}/e.$$

## 3. Results and Discussion

For studying the decay of the excitations of the dipoles in the chain let’s change its length and the time of the external field action upon it. Let us specify the size r=10−100 nm of nano dipoles with spacing *a* in the chain as well as the parameters of the substrate surrounding the oscillators and the frequency of the alternating field.

The model calculations show the changes in the number of the uncompensated charges (cations and anions) $c_{n}$ with the mass *m* in the shells at the ends of the dipoles. The changes in the number of the charges $c_{n}$ show the character of the growth or decay of the dipole excitations in the chain. In the present investigation the number *n* varies in the range of 1–30. The boundary conditions for the chain of the oscillators of l=2na in length are standard as in our previous investigation [23]. Let us set the parameters for model equations: the strength of the external field is $E=10^{-0}-10^{-5}$; the dipole radii are r=10−100 nm; the relaxation time of bound charges in the shells is $\tau =1.6\times 10^{-5}$ s; for dielectric permeability the maximum low-frequency value is ε=650 and minimum high-frequency value is $\varepsilon_{\infty }=8$; the number of charges at the ends of the dipoles is β=1 induced in the electric field with unit strength; the dissipation parameter is $\xi =10^{-0}$, and $m=1.6\times 10^{-27}$ kg. For heterogeneous systems in the nanoscale region the most interesting characteristics of responses are at the low frequencies $\Omega_{1}=8$ Hz, $\Omega_{2}=13$ Hz and $\Omega_{3}=20$ Hz. The parameters used in the model equations are presented in Table 1. Figure 1 shows n=3 dipoles (with radii r) with spacing *a* in the model chain.

The strongest excitation in the chains of the dipoles should be reached at the smallest distance 2a=r between them at any intensity of the external electric fields action on them. The increase of the distance 2a>r leads to the acceleration of the decrease in the level of the excitation of the dipoles in the chains. This seems to be natural because of the binding forces of the charges are inversly proportional to the distance between them.

The numerical calculations for the above systems show that after the cessation of the action of strong or weak alternating field (the fields were switched off programmatically to the value of E=0 V/m for times $t\geq 1\times 10^{-3}$ ) at any low frequencies the excitations in the chains do decay. The longer is the chain, the longer is the time of decay.

However, the dependence of the decay time on the chain length has a non-linear character as it is shown in Figure 2a–c. For the strength of the external alternating fields $E=10^{-5}$ V/m, the dependences of the number of charges at the ends of the dipoles on the lengths of the chains *n* in the range of 1–30 and the time of the external decay at the frequency Ω=13 Hz at the dipole radii r=10 nm with spacing 2a=r are presented in Figure 2a; for the strength of external fields $E=10^{0}$ V/m the dependences are shown in Figure 2b; and for the strength of the external fields $E=10^{5}$ V/m they are given in Figure 2c.

Among the peculiarities of the decay character the following should be noted. When the external action on the chains of dipoles with variable moments is switched off, the effect of decay occurs.

After the switching off of the strong fields of $E=10^{1}÷10^{6}$ V/m the decay has similar duration at any length of the dipole chains and tends to 0 starting with time t≈1 s and ending at $t\approx 10^{2}$ s. At some weak fields $E=10^{-6}÷10^{0}$ V/m the decay of short dipole chains, n≤ 19–20, the time of decay is very small, about $t\approx 10^{-3}$ s. However, starting with the length of n≥20, the decay time sharply increases by five orders of magnitude, up to $t\approx 10^{2}$ s. The characteristics of the decay of the excited chain experiencing super-weak and strong actions are similar, but without the area which is present in the chains with the length n≤19÷20. Let us also note that at any fields acting on the chains, the maxima of the excitations in the dipoles have almost equal values as it should be in the systems of dipoles with variable moments [23]. This due to that the maxima of the excitations in the chains are limited by the number of mobile charges in the electric double layers of the considered dipoles. For the limiting case when there is only one mobile charge in the dipole double layer, the characteristics of the excitations and decay is shown in Figure 3. It is seen that there is very low amplitude of the excitations in such chain.

In the chains with the number of dipoles n≤19−20 and only one mobile charge in each dipole the excitations are completely absent. At the same parameters that are used for the chains with the dipole radii of $r\approx 10^{-8}\;$ m, the similar strong excitation is observed in the chains with $r\approx 10^{-6}\;$ m, which is, however, larger by two orders of magnitude. Let us note here that in the present calculations the possibility of the influence of the thermodynamic parameters [21,28] and electrostatic screening of surrounding substrate [29] on the length of the chains has not been taken into account.

## 4. Conclusions

Thus, in the present paper, the character of the decay of the excitations in the chains of dipoles with variable moments is established after the external alternating electric field has been switched off at biologically significant low frequency.

The numerical calculations show that the decay of the excitation of electric dipoles occurs after the low-frequency field of any intensity has been switched off. At strong low-frequency fields with $E=10^{1}÷10^{6}$ V/m the dipole excitations are decaying for a long time up to $t\approx 10^{2}$ at any length of chains. At weak fields with $E=10^{-6}-10^{0}$ V/m, it is found that the excitations of the dipoles in the short chains with n<19−20 decay five orders of magnitude faster than in the long chains with n≥19÷20. At super weak external electric fields, the decay of the excitations of dipoles is similar to that in the dipole chains in weak fields, however, the excitations and decays of the short chains are absent.

At growing frequency of the external field acting on the chain, the characteristics of the excitation decay changes in the way similar to that observed at the increase of the strength of the external field. 

## Figures and Tables

**Figure 1 nanomaterials-11-00074-f001:**
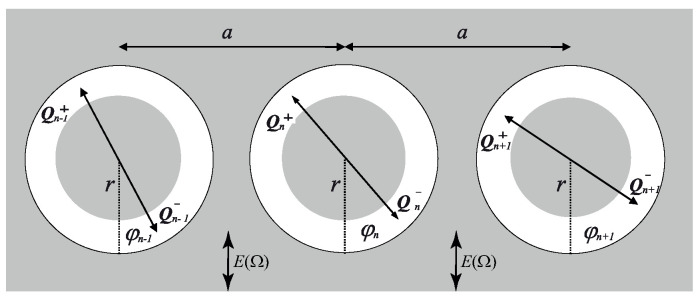
Schematic representation of the model system in the form of the chain of non-point dipole oscillators of diameter 2r with spacing *a* where the oscillations of the charges in the shells (the real thickness of which can be about ∼10–100 ) around nanoscale particles are shown. The direction of the action of the alternating electric field E(Ω) is denoted by arrows.

**Figure 2 nanomaterials-11-00074-f002:**
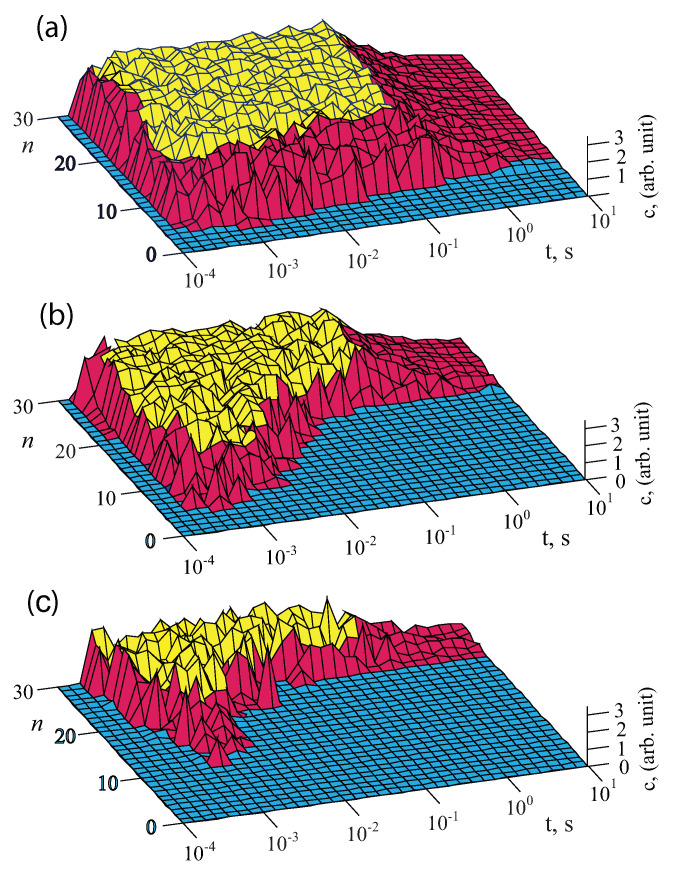
Dependences of the number of charges at the ends of non-point dipoles on the change of the time of the external excitation at the frequency Ω=13 Hz and the length of the chain with dipole radius r=10 nm and spacing 2a=r for the strengths of the external alternating fields $E=10^{5}$ V/m (**a**), $E=10^{0}$ V/m (**b**) and $E=10^{-6}$ V/m (**c**). The excitation field was switched off at the instant of time $t\geq 1\times 10^{-3}$ s.

**Figure 3 nanomaterials-11-00074-f003:**
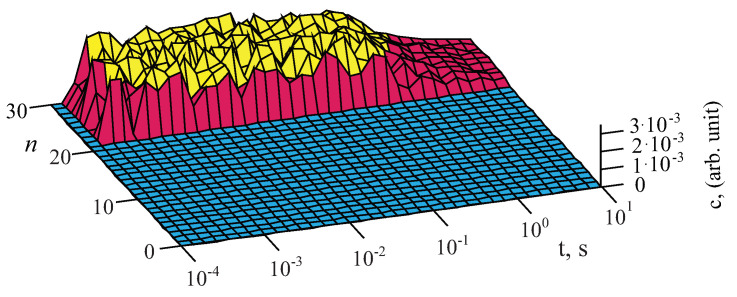
Dependences of the number of the charges at the ends of non-point dipoles on the changes of the length of the chains for the same parameters that are in Figure 2 but at the minimal number of charges $c_{n}=1$. The excitation field was switched off at the instant of time $t\geq 1\times 10^{-3}\;s.$

**Table 1 nanomaterials-11-00074-t001:** Parameters used in the model.

Parameter	Value
Number of uncompensated charges	$c_{n}=100$
Size of nanoscale dipoles	r=20 nm
Number of dipoles in chain	*n* from 1 to 30
Dipoles spacing in chain	*a* (2a>r) nm
Proton mass	$m=1.67\times 10^{-27}$ kg
Strength of external low-frequency field	$E=10^{-5}-10^{5}$ V/m
Frequency of external field	$\Omega_{1}=13$ Hz
Relaxation time of bound charges	$\tau =1.6\times 10^{-5}$ s
Dissipation parameter	$\xi =10^{-0}$
High-frequency dielectric permeability	$\varepsilon_{\infty }=8$ F/m
Low-frequency dielectric permeability	$\varepsilon_{s}=650$ F/m

## Data Availability

The data presented in this study are available on request from the author.

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
