# Peer review of "Electrical Excitation Decay Time in Chains of Nanoscale Non-Point Dipoles"

_nanomaterials, 2020, doi:10.3390/nano11010074_

Round 1

Reviewer 1 Report

Based on the model developed in the works below:

[1] Evgeniy G. Fateev, Supersensitivity in a chain of closely spaced electric dipoles with variable moments, Physical Review E, Volume 65, 021403 (2002), DOI: 10.1103/PhysRevE.65.021403

[2] E. G. Fateev, Chains of nanoscale dipoles in alternating electric fields, Technical Physics Letters, 2018, Vol. 44, No. 8, pp. 655–658 (2018), DOI: 10.1134/S1063785018080060

, the author studies the electrical excitation decay time in chains of nanoscale non-point dipoles.

The approached topic is interesting and topical for the field of nanosciences.

Abstract- covers the essence of the paper well.

Introduction - is short but substantial and clear presented.

The model system is the same with the one presented in the articles listed above.

Results and discussion are sufficient and the explanation is reasonable.

Conclusions: the results of the study are synthesized in concordance with work.

This paper also cites suitable references, in comprehensive orders.

Author Response

Sent to reviewer 1.

Reviewer 2 Report

In the manuscript, a chain of interacting non-point dipole oscillators is investigated using the model and the method developed in the preceding publication by the author, Ref. 22. The present work is aimed to calculate the decay of the dipole chain.

The obtained results are interesting and deserve publication. I have a minor remark. The decay appears in the problem due to introduced phenomenological dissipative forces, and therefore the dissipation parameter is a fitting one. It would be appropriate to add a brief discussion regarding fitting parameters in the developed approach on their influence of the predictive power of the theory.

In my opinion, the paper can be published in Nanomaterials encouraging the authors to consider the aforesaid suggestion.

Author Response

Sent to reviewer 2.

Reviewer 3 Report

The present work is the further development of previous publications by the author (references 22-23). However, it is not enough clear, which are the improvements  and the new achievements...  the manuscript can be substantially improved, if the comparison with previious results is  better discussed.

The abstract is also confusing...

There are some typos...

Author Response

Sent to reviewer 3.

Round 2

Reviewer 3 Report

The author_response is  very cryptic...

It is said: "The model focuses on the damping of excitations and damping times for systems with chains of different lengths. In general, for the first time it was established with what speed such excitations are damped. This is important for many applications of the physics of nanomaterials. Line 1-3"

But this text does not appear in the manusript. It should  be  nice to introduce it...

It is mentioned:  "Figures 2a, 2b, 2c."

But   I  do not see any changes in these figures...

"Lines 1-134."  What does the author mean? I do not understand...

The  abstract had been changed.... OK!

After equation (5) a  short paragraph  has  been introduced  about the dissipative parameter.

A  new reference has been introduced [27].

In general, the changes are not substantial. I  think, actually,  it can be better improved.

Author Response

The abstract has been edited to make the purpose and results of the article easier to understand.

The changes in the abstract emphasize the importance of this article and its differences from the other articles of the author.

The model focuses on the decay of excitations and the decay times for the systems with the chains of different lengths. Specifically, the rate of decay of such excitations has been established for the first time. This is important for many applications of the physics of nanomaterials.

Line 1-3.  Lines 70-80, 90-92, 126-127.  Figures 2a, 2b, 2c. See file fateev old.pdf

The manuscript has been proofread with a number of typographical errors fixed.

Lines 1-134. See file fateev old .pdf

In version 1 of the article, the changes to the text of the manuscript are marked in the PDF file. There are about 40 such places. The article added to the bibliography is necessary, as the arguments important for this article follow from it.
